# Osmotic Stress Interferes with DNA Damage Response and H2AX Phosphorylation in Human Keratinocytes

**DOI:** 10.3390/cells11060959

**Published:** 2022-03-11

**Authors:** Laura Hoen, Christoph Rudisch, Michael Wick, Daniela Indenbirken, Adam Grundhoff, Florian Wegwitz, Stefan Kalkhof, Janosch Hildebrand

**Affiliations:** 1Institute for Bioanalysis, Coburg University of Applied Sciences and Arts Coburg, Friedrich-Streib-Straße 2, D-96450 Coburg, Germany; laura.hoen@hs-coburg.de (L.H.); c.rudisch@web.de (C.R.); michael.wick@hs-coburg.de (M.W.); stefan.kalkhof@hs-coburg.de (S.K.); 2Proteomics Unit, Leibniz Institute for Experimental Virology (HPI), Martinistraße 52 (N63), D-20251 Hamburg, Germany; daniela.indenbirken@leibniz-hpi.de (D.I.); adam.grundhoff@leibniz-hpi.de (A.G.); 3Department of Gynecology and Obstetrics, University Medical Center Göttingen, Robert-Koch-Straße 40, D-37075 Göttingen, Germany; fwegwit@gwdg.de; 4Fraunhofer Institute for Cell Therapy and Immunology, Perlickstraße 1, D-04103 Leipzig, Germany

**Keywords:** keratinocytes, UV radiation, gene regulation, DNA repair disorders, osmotic stress

## Abstract

The human skin and in particular its outermost layer, the epidermis, protects the body from potentially harmful substances, radiation as well as excessive water loss. However, the interference between the various stress responses of the epidermal keratinocytes, which often occur simultaneously, is largely unknown. The focus of this study was to investigate the interference between osmotic stress and DNA damage response. In addition to revealing the already well-described regulation of diverse gene sets, for example, cellular processes such as transcription, translation, and metabolic pathways (e.g., the KEGG citrate cycle and Reactome G2/M checkpoints), gene expression analysis of osmotically stressed keratinocytes revealed an influence on the transcription of genes also related to UV-induced DNA damage response. A gene network regulating the H2AX phosphorylation was identified to be regulated by osmotic stress. To analyze and test the interference between osmotic stress and DNA damage response, which can be triggered by UV stress on the one hand and oxidative stress on the other, in more detail, primary human keratinocytes were cultured under osmotic stress conditions and subsequently exposed to UV light and H_2_O_2_, respectively. γH2AX measurements revealed lower γH2AX levels in cells previously cultured under osmotic stress conditions.

## 1. Introduction

The human skin consists of a multilayered epithelium, which is the barrier between the environment and the organism [1]. The epidermis, largely consisting of keratinocytes [2], serves as the principal barrier and protects the body from the penetration of pollutants and ultraviolet (UV) radiation, and also prevents water loss [3]. One key factor for healthy skin is efficient DNA repair, for example, in order to prevent skin cancer [4].

The condition of skin can be affected by environmental conditions, e.g., humidity [5], or by endogenous factors, e.g., dehydration, which leads to reduced skin turgor [6] or poor wound healing [7]. These signs indicate that the skin and skin cells are affected by the intracellular body water level. One form of dehydration is hypertonic dehydration, which is often due to inadequate water intake. It is defined by an increased sodium concentration in the extracellular fluid (>300 mOsm/kg) [8], whereby the osmotic pressure on the cells rises and osmosis leads to cell shrinkage and the intracellular homeostasis of inorganic ions. The normal sodium concentration in blood serum is between 135–145 mmol/L [9]. This can increase due to a loss of water [10] and can consequently lead to an osmotic stress response. Besides osmotic stress, UV radiation is one major environmental factor affecting skin cell physiology [11]. Skin cells induce DNA damage response to repair UV-induced DNA damages such as double-strand breaks [12]. Alfieri et al. pointed out similarities between different stress responses, such as hyperosmotic and UV stress responses [13]. Human skin cells exposed to osmotic stress regulate heat shock protein (HSP) 70 and HSP27 by p38 mitogen-activated kinase (MAPK) p38 [14]. MAPKs are a well-studied osmotic stress signaling pathway in mammalian cells [15]. Diverse extracellular stimuli like hormones, UV irradiation, and osmotic stress activate MAPKs. This activation leads to multiple fundamental cellular processes such as proliferation, growth, cell cycle control, and apoptosis [15,16]. The p38 MAPK signaling pathway can be induced in response to DNA damage occurring after osmotic stress [17,18]. There are hints that the activation of p38 by ataxia–telangiectasia mutated (ATM) may act as an alternative to p53, as a cell cycle checkpoint pathway involving p38 and MK2 lead to cell cycle arrest and survival after DNA damage in p53-deficient cells [19]. ATM and ATR (ATM and Rad 3-related) are the main kinases involved in DNA damage repair [20,21]. ATM senses double-stranded breaks, whereas ATR predominantly senses single-stranded breaks. These checkpoint kinases transfer signals to effector molecules and phosphorylate histone family member X (H2AX), which is the key event in DNA damage response [20,22]. The phosphorylation of H2AX at Ser139, which is called γH2AX, is one of the early cellular responses to DNA double-strand breaks (DSBs) [23], which makes it a useful biomarker for DSBs [24]. Osmotic stress induced by NaCl can induce a cell cycle arrest in a similar manner to DNA damage by UV radiation [25,26]. In fact, increased DNA double-strand breaks were reported after osmotic stress in murine kidney cells [27], which persisted even when cells adapted to the osmotic stress [28].

This shows that cellular stress pathways form complex networks and that the interference between different stress responses is of major importance, as it could also help to develop new therapeutic targets [29]. However, the interference between osmotic stress and DNA damage response in human skin cells is largely unknown. In this study, we investigate the interference between osmotic stress and DNA damage response in primary human skin keratinocytes. For this purpose, the transcriptional response to osmotic stress was comprehensively investigated using high-throughput transcriptome analyses, revealing interference between osmotic stress and DNA damage response. The interference was validated by γH2AX measurements in simultaneous stress treatment, and a functional gene network controlling phosphorylation of H2AX was identified to be affected by osmotic stress, explaining the interference between these stress responses in human keratinocytes.

## 2. Results

### 2.1. Characterization of Gene Expression Changes upon Osmotic Stress in Primary Human Keratinocytes Revealed Interference between Osmotic Stress and DNA Damage Response

To investigate gene expression changes after osmotic stress, human primary keratinocytes were cultivated under osmotic stress conditions. For this purpose, the media was supplemented with 200 mM NaCl for 3, 6, 12, and 24 h. The activation of osmotic stress response was validated by western blot analysis of p38 phosphorylation. Densitometric analysis revealed a strong p38 activation after 3 and 6 h, which declined over time after 24 h (Appendix A). To investigate the early and midterm transcriptional response upon osmotic stress, transcriptomic analyses were performed from RNA extracts of three biological replicates after treatment with 200 mM NaCl for 6 and 12 h.

Principal component analysis and clustering by heatmaps showed the separation of untreated keratinocytes and cells cultured under hyperosmotic conditions for 6 and 12 h, indicating significant transcriptional changes during osmotic stress (Appendix A). Although one replicate of the 12 h treatment showed a slightly different response, which is evidenced by the clustering, all the replicates were included in the analysis for statistical relevance. When compared to untreated cells, of the identified genes, 1154 genes were significantly regulated after 6 h of NaCl treatment (360 upregulated, 794 downregulated, Figure 1A) and 1183 genes underwent significant changes after 12 h (360 upregulated, 794 downregulated; Figure 1B).

Gene set enrichment analysis was performed to identify regulated gene sets and perturbed cellular functions between untreated keratinocytes and cells cultured under hyperosmotic conditions. For this purpose, the C2 collection of curated gene sets was used, which is representative of canonical pathways from diverse pathway resources [30]. Keratinocytes cultured under hyperosmotic conditions for 6 h enriched 2489 gene sets, whereas cells treated for 12 h enriched 1983 gene sets. A network analysis using the Cytoscape plugin based on the normalized enrichment score (NES) of the enriched gene sets was performed to identify positively and negatively correlated gene sets after 6 and 12 h of treatment (Figure 2, FDR < 25%, *p* < 0.05). Most positively correlated gene sets affected general processes such as transcription, translation, and the citric cycle. Interestingly, several gene sets involved in the UV and DNA damage response of keratinocytes and XP model systems were negatively correlated.

The dazard gene set [31] reflects the UV response in keratinocytes and showed negative NES values of −2.02 and −2.16 in cells cultured under hyperosmotic conditions for 6 and 12 h, respectively (Appendix A). The dacosta gene set [32] characterizes DNA damage response in XP models and revealed NES values of −1.85 and −2.22 in cells cultured under osmotic stress conditions for 6 and 12 h, respectively (Appendix A).

These findings were in line with the assumption that gene expression changes during osmotic stress and interfere with UV and DNA damage response in human keratinocytes. A detailed analysis of the respective gene sets suggested that a gene network controlling the phosphorylation of H2AX, a key event during DNA damage repair, is affected by osmotic stress (Figure 3).

The phosphorylation of H2AX is mediated by PIKK family proteins, including ATM, ATR, and DNA dependent protein kinase catalytic subunit (DNA-PKcs). ATM and DNA-PKcs are part of the DSB signaling cascade [33,34], whereas ATR was shown to phosphorylate H2AX during replication fork blockage induced by UV exposure [35]. The log2FC of ATM and ATR showed negative regulation at both time points upon induction of osmotic stress, but was only significant after 12 h (Figure 3). The dephosphorylation of γH2AX is mediated by phosphatases such as protein phosphatase 2A (PP2A) or protein phosphatase 4 (PP4). Here, ATM was also involved by phosphorylating coiled-coil domain-containing 6 (CCDC6), which negatively regulates the activity of PP4 [36]. CCDC6 was found to be significantly downregulated at both time points (*p* < 0.0005), whereas PP4 was significantly upregulated in cells under hyperosmotic stress conditions at both time points (*p* < 0.005). Taken together, the regulation of this gene network indicates that osmotic stress interferes with H2AX phosphorylation and DNA damage response in human keratinocytes. qPCR analysis of ATM, ATR, CCDC6 and PP4 from keratinocytes treated with 200 mM NaCl for 6 hours (Appendix A) validated the sequencing data (Figure 3).

**Figure 3 cells-11-00959-f003:**
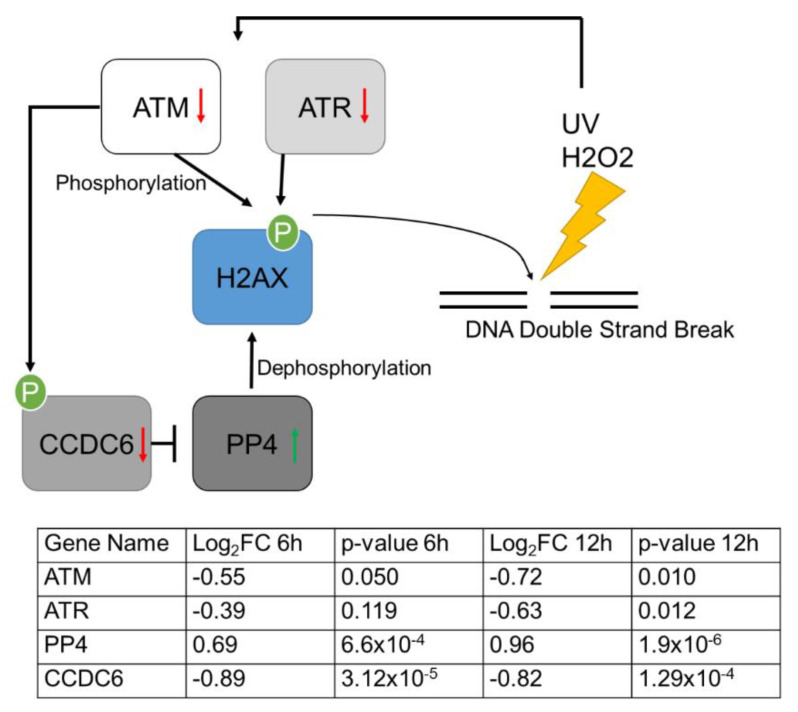
Gene network affected by osmotic stress regulating H2AX phosphorylation (modulated after [37]). H2AX phosphorylation is a key event during DNA damage repair. ATM and ATR phosphorylate H2AX upon DNA damage. Dephosphorylation is controlled by PP4, which is inhibited by CCDC6. Arrows indicate gene expression changes after osmotic stress. The table shows the log2 fold changes and *p*-values of respective genes after cultivation under osmotic stress conditions for 6 and 12 h (*n* = 3).

### 2.2. Costress Model of Osmotic Stress and UV Radiation Revealed Decreased Ability of DNA Damage Repair via Phosphorylation of H2AX

To validate this hypothesis, costress models using primary human keratinocytes were established. Cells were cultured under normal or hyperosmotic conditions for 6 h and were subsequently cultured under normal conditions or were irradiated by sun-simulated radiation or treated with H_2_O_2_ to induce DNA damage. To monitor the interference of osmotic stress with H2AX phosphorylation, the γH2AX levels were measured 3 h after irradiation or H_2_O_2_ treatment by immunofluorescence staining and image analysis. This double stress could be potentially fatal for the cells, but the survival of the cell is crucial for the purpose of immunofluorescence staining (see Appendix A). The increase of the γH2AX level was calculated as the fold change of γH2AX positive cells, comparing irradiated or H_2_O_2_ treated with untreated cells. The percentages of γH2AX positive cells for each experiment, including significant outliers, show a difference in the basic values (Appendix A), but the increase in the different treatments is crucial, and therefore the fold change compared to the respective controls was calculated.

Cells cultured under normal conditions before DNA damage induction revealed a 2.4 fold increase of γH2AX positive cells after radiation and a 4.7 fold increase after H_2_O_2_ treatment (Figure 4). Cells cultured under hyperosmotic conditions before DNA damage induction revealed fold changes of γH2AX positive cells of only 1.2 or 1.9 after radiation or H_2_O_2_ treatment, respectively.

This finding is in line with the assumption that osmotic stress interferes with DNA damage response via influencing H2AX phosphorylation in human keratinocytes and shows the importance of the interference between different stress responses in human skin cells.

## 3. Discussion

This study revealed an interference of osmotic stress and UV or DNA damage response in human primary keratinocytes.

The transcriptome analysis via NGS of the early and midterm osmotic stress response of human keratinocytes identified hundreds of differentially expressed genes. In line with the p38 phosphorylation, more differentially expressed genes were found 6 h after the beginning of osmotic stress than after 12 h. This shows the transient nature of the osmotic stress response and could also indicate adaption processes to osmotic stress [13].

As expected, gene set enrichment analysis revealed that gene sets associated with general cellular processes, such transcription, translation, and metabolic pathways (e.g., citric cycle), were upregulated [38,39,40,41]. Interestingly, we identified two downregulated gene sets associated with UV and DNA damage response, indicating interference between cell processes involved in UV and DNA damage response and osmotic stress. In line with this, Kültz et al. investigated a transient cell cycle arrest after acute elevation of NaCl in cell culture [18]. This cell cycle arrest was similar to that observed when cells were exposed to ionizing or ultraviolet radiation. In both cases, the arrest occurs within minutes and is maintained for hours before the cells begin to proliferate again [25,26], underscoring the transient nature of osmotic stress response. Different studies in non-skin cells have shown that osmotic stress induces DNA damage [25,26]. Nevertheless, as these studies were performed on other cell types, their results cannot be extrapolated to skin keratinocytes. Despite this, Dmitrieva et al. showed that high NaCl impairs DNA damage repair after UV and IR in mouse skin cells. Mre11, a component of the MRN complex, which plays a central role in double-strand break repair, leaves the nucleus under high NaCl concentrations, which leads to decreased H2AX phosphorylation. Mre11 normally detects DNA damage in the nucleus and activates DNA damage repair by ATM and ATR [42]. However, Mre11 was found not to be significant in the transcriptomic data because it cannot differentiate between the nucleus and the cytoplasm. Our study revealed the regulation of a gene network, including ATM and ATR, leading to decreased phosphorylation of H2AX. In combination with the observed decrease of γH2AX during DNA damage response in osmotically stressed cells, this allows for the assumption that osmotic stress leads to extenuated DNA damage response or to a time shift of key events, such as the phosphorylation of H2AX. The phosphorylation of H2AX might not only be influenced by DNA damage. Serum starvation also leads to γH2AX via the p38 MAPK signaling pathway [42,43], which is also activated by osmotic stress [14,15].

The identified gene network focused on ATM and ATR, which are activated upon DNA damage [35]. Both kinases were downregulated upon osmotic stress in keratinocytes (Figure 3 and Appendix A). ATM senses DNA double-strand breaks and is important for UV-induced DNA damage repair. Its downregulation upon osmotic stress can interfere with any subsequent UV damage repair response. This interference was validated using H_2_O_2_ treatment as a DNA damaging agent. Hydrogen peroxide induces the phosphorylation of H2AX mediated by ATR, whereby DNA double-strand breaks are not required [44]. Therefore, the effect on phosphorylation of H2AX during H_2_O_2_ treatment can be independent of DNA double-strand breaks, but also relies on extenuated ATR activation. In addition, some differences in the responses of keratinocytes to stress induced by H_2_O_2_ or UVA radiation have been reported. Hydrogen peroxide caused a pronounced G2/M cell cycle arrest, whereas UVA radiation did not affect cell cycle progression [45]. Cell cycle arrest occurred 24 h after H_2_O_2_ treatment, so this time point was not considered in this study. However, Thorn et al. showed that the stress response to H_2_O_2_ is different from that to radiation [45], which should be considered when comparing results. As Staszewski et al. showed, there is a time-dependent foci formation after treatment with UV-C or alkylating agents. The maximum γH2AX foci after UV-C treatment was reached 12 h after treatment, whereas the peak was reached 8 h after treatment with alkylating agents [46]. The decrease in γH2AX foci after this time point should not have influenced the measurements, as γH2AX was measured 3 h after irradiation or H_2_O_2_ treatment. It is known that DNA repair deficits can lead to carcinogenesis [47]. ATM and ATR in particular show high potential for use as therapeutic targets [48]. Also, in skin cancer, more specifically in Merkel cell carcinoma (MCC), a highly aggressive skin cancer which is associated with chronic exposure to UV [49], 29% of patients showed abnormalities in DNA repair genes, such as ATM [50]. Protein phosphatase 4 is one of four known phosphatases known for the dephosphorylation of γH2AX [37], which was found to be upregulated by osmotic stress. As osmotic stress is a major consequence of dehydration, this study implies consequences for dehydration and skin conditions, such as dry skin or skin cancer [47,49,50]. Dehydration, not only due to heavy water loss, is a major problem, especially in the elderly [6,7,8]. Hypertonic dehydration is often a result of inadequate water intake [8]. This is consistent with the stress model shown here, where higher NaCl concentrations are responsible for osmotic stress. As pointed out, osmotic stress interferes with the ability for an adequate response to UV-induced DNA double-strand breaks, which can lead to skin cancer [4,51,52]. Population-based studies claim decreased DNA repair capacity during aging [53]. A recent study revealed that the DNA repair ability via the base excision repair (BER) factors in aged keratinocytes is decreased [54], which was shown to be activated by ATM-p53 signaling in HepG2 cells [55].

Taken together, this study shows the first hint of a connection between osmotic stress and DNA damage responses in human keratinocytes. In fact, osmotic stress impairs the ability of the DNA damage response mechanism via γH2AX in human keratinocytes. As a result, our study suggests that dehydrated skin cannot respond as efficiently as normal skin to UV radiation and DNA damage.

## 4. Materials and Methods

### 4.1. Cell Culture

Primary normal human epidermal keratinocytes provided by Lonza were cultured in KGM Gold Keratinocyte Growth Medium BulletKit (Lonza, Basel, Switzerland) at 37 °C with 5% CO_2_. Three days after seeding cells to 6-well plates (100,000 cells/well), the cells were treated with sodium chloride (NaCl, Carl Roth, Karlsruhe, Germany) in the given concentration in the medium or were simply given a media change (control).

### 4.2. RNA Isolation and Sequencing

For RNA isolation, cells were trypsinized and collected in PBS (Roth) with 10% FCS (Biowest, Nuaillé, France). After centrifugation for 5 min, 500 *g* at room temperature, the supernatant was removed and either the total RNA was isolated directly with the mirVana RNA Isolation Kit (Thermo Fisher Scientific, Darmstadt, Germany) according to the manufacturer’s protocol or the pellet was frozen at −80 °C and RNA isolation wassperformed afterwarda.

For total RNA sequencing, RNA integrity was estimated with the RNA 6000 Nano Chip on an Agilent 2100 Bioanalyzer (Agilent Technologies, Santa Clara, CA, USA). From total RNA, mRNA was extracted using the NEBNext Poly(A) mRNA Magnetic Isolation module (New England Biolabs, Frankfurt am Main, Germany) and RNA-Seq libraries were generated using the NEXTFLEX Rapid Directional qRNA-Seq Kit (Bioo Scientific, Austin, TX, USA) as per the manufacturer’s recommendations. Concentrations of all samples were measured with a Qubit 2.0 Fluorometer (Thermo Fisher Scientific, Darmstadt, Germany) and fragment lengths distribution of the final libraries was analyzed with the DNA High Sensitivity Chip on an Agilent 2100 Bioanalyzer (Agilent Technologies). All samples were normalized to 2 nM and pooled equimolar. The library pool was sequenced on the NextSeq500 (Illumina, Berlin, Germany) with 1 × 75 bp, with 21 to 24 mio reads per sample. Raw data related to this article can be found at http://www.ncbi.nlm.nih.gov/bioproject/773592, release date: 01/07/2022, hosted by the Sequence Read Archive (SRA).

### 4.3. Analysis of Sequencing Data

Sequencing data was analyzed using the Galaxy Platform (www.usegalaxy.eu/access date 20/08/2021). Raw sequencing data were trimmed with FastQ Trimmer version 1.1.5 [56] 24 bases from the 5′end to remove low-quality regions (assessed with FastQC). Reads mapping to the hg38 was performed by RNA STAR version 2.7.5b [57]. Counts of each gene were generated by featureCounts [58] (versions: subread (1.6.4), samtools (1.9), coreutils (8.31)) and normalization was performed using DESeq2 version 1.22.1 [59]. Normalized counts were further analyzed with Gene Set Enrichment Analysis (GSEA), version 4.1.0 [60], with the following settings: gene matrix: c2.all.v7.4.symbols.gmt (Gene ontology); gene set size filter: 15 (minimum) and 5000 (maximum); false discovery rate (FDR) of <25% and *p* < 0.05. A total of 4258 gene sets successfully passed these criteria. Network of gene set enrichment analysis was visualized with Cytoscape v3.8.0. Plots were either generated by Galaxy or separately made by R with R Studio version 1.3.1093 with ggplot2 [61].

### 4.4. qPCR (TaqMan Assays)

After RNA Isolation, RNA was synthesized to complementary DNA (cDNA) with the High-Capacity RNA-to-cDNA™ Kit (Thermo Fischer, Darmstadt, Germany) according to the manufacturer’s protocol. This was performed on a Thermal Cycler T100 (BIO-RAD Labaratories, Hercules, CA, USA). A uniform RNA concentration was used, starting with the lowest.

Afterwards, the cDNA was used for the TaqMan Assays (Thermo Fischer, Darmstadt, Germany). The following assays were performed according to the manufacturer’s protocol, each in double determination for each probe.

ATM (Assay ID: Hs00175892_m1), ATR (Assay ID: Hs00992123_m1), PP4 (Assay ID: Hs00908713_m1), CCDC6 (Assay ID: Hs00193731_m1), 18S (Assay ID: Hs99999901_s1).

The assays were performed on a qPCR thermal cycler C1000 fitted with a CFX96 Real-Time System (BIO-RAD Laboratories, Hercules, California, USA). Analysis were performed using Microsoft Excel 2016. First, the C_q_ values were normalized to the 18S level (C_q_ of Assay x minus C_q_ of 18S Assay). Further analysis was done according to Schmittgen et al. [62]. Significant outliers (according to Grubbs Test) were excluded.

### 4.5. Costress Experiments by Osmotic Stress and Either UV Irradiation or Oxidative Stress (H_2_O_2_ Treatment)

Cells were equally seeded in 96-well plates and incubated for 3 days. Afterwards, they were either treated with NaCl-enriched medium (100 mM or 200 mM) or given a medium change (control). The cells were irradiated in a Q-SUN Xe-1 (Q-Lab, Saarbrücken, Germany) with Daylight Q Filter for 10 min with 75 W (4500 mJ/cm^2^) 6 h later. One well for each treatment was irradiated, while the other was covered with aluminum foil. In both of them, the medium was replaced by PBS prior to irradiation. After irradiation, cells were incubated in medium again for 3 h. Replicates were made biologically and technically on different days on different 96-well plates. In a similar fashion to the irradiation treatment, cells were treated with 300 µM hydrogen peroxide (H_2_O_2_) (Thermo Fisher Scientific, Darmstadt, Germany) in the medium for 10 min after incubation with 100 or 200 mM NaCl for 6 h. After treatment with H_2_O_2_, cells were incubated in medium again for 3 h. The experimental setup is shown in Figure 4B.

### 4.6. γH2AX Staining and Image Analysis

After the incubation phase, cells were stained according to [63] with the following changes: after fixation, cells were permeabilized for 10 min with 0.1% Triton X-100 (Merck, Darmstadt, Germany) in PBS and then washed. Blocking was done with 0.5% BSA (Carl Roth, Karlsruhe, Germany) in PBS for 30 min. All used antibodies were diluted in 0.5% BSA in PBS. The primary antibody was P-Histone H2A.X (S139) (γH2AX) Lot 14 (Cell Signaling, Frankfurt am Main, Germany), the secondary antibody was Alexa Fluor^TM^ 555 goat anti-rabbit IgG (Lot 2018130, Thermo Fisher Scientific, Darmstadt, Germany), and the third was DAPI (Lot TD2559303, Thermo Fisher Scientific, Darmstadt, Germany). After staining, cells were kept in PBS (Carl Roth, Karlsruhe, Germany). Fluorescence images were taken using a Zeiss Axio Observer.Z1 (Carl Zeiss, Oberkochen, Germany) with the software Zen, 2012. Identification of γH2AX positive cells was achieved via the software Wolfram Mathematica 12 (version 12.0.0.0) and ImageJ (version 1.52A, Wayne Rasband, USA). DAPI images were used to crop the cells from fluorescent images of γH2AX. Then, a background subtraction was performed, and each cell was given the brightest red pixel for the whole cell. The optimized images were subsequently analyzed with ImageJ version 1.52a. By setting an appropriate threshold for the untreated control of one experiment, which then was used for all other images of this experiment, the amount of γH2AX positive cells in percent were defined. Out of these percentages, the fold change between treated (either irradiated or H_2_O_2_) and untreated cells in control and osmotically stressed cells was calculated. After testing for normal distribution, a paired *t*-test was performed to test for significance. Significant outliers (according to Grubbs Test) were excluded.

## Figures and Tables

**Figure 1 cells-11-00959-f001:**
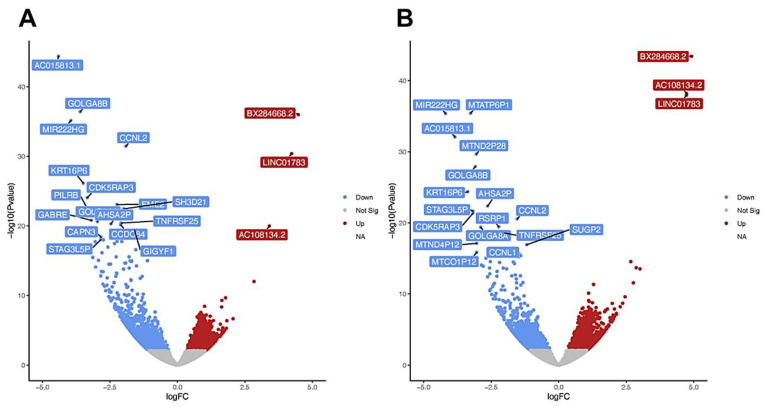
Volcano Plots of transcriptome analysis of keratinocytes treated with 200 mM NaCl for 6 h (**A**) and 12 h (**B**) (each *n* = 3). Each point represents an individual gene. The y-axis shows –log_10_(*p*-value) for the associations per gene, and the x-axis shows the effect, log_2_FC. Grey represents non-significant genes (*p* > 0.05), blue downregulated, and red upregulated.

**Figure 2 cells-11-00959-f002:**
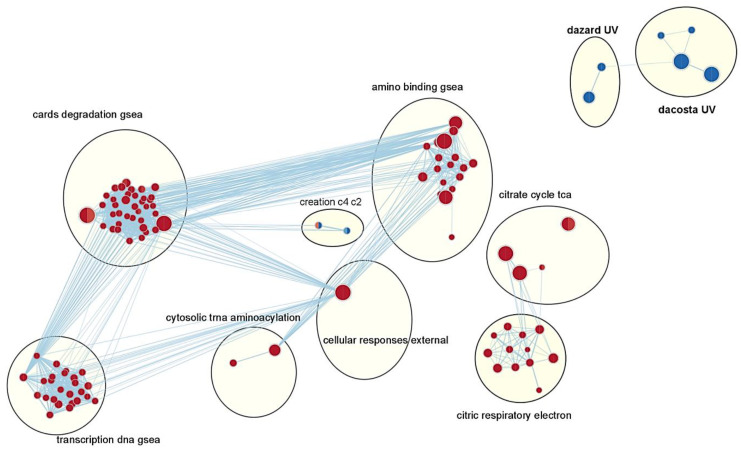
Network analysis of gene set enrichment. Each node represents one gene set. The left half of the circle represents the correlation after 6 h of NaCl treatment, and the right half of the circle represents the correlation after 12 h of NaCl treatment (each *n* = 3). The size of the nodes represents the number of genes in the node; edges represent the correlation of the nodes; *p* = 0.05, red = positively correlated gene sets, blue = negatively correlated gene sets, bold: gene sets of interest; nodes with no edges to other gene sets are not shown.

**Figure 4 cells-11-00959-f004:**
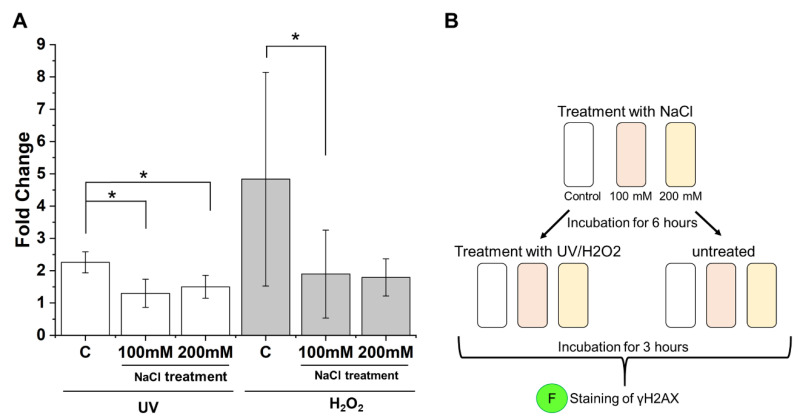
γH2AX identification in costress model. (**A**) Fold change of γH2AX positive cells after UV radiation (*n* = 4) or H_2_O_2_ treatment (*n* = 8) in untreated control cells or cells cultured under osmotic stress conditions prior to radiation or H_2_O_2_ treatment. Fold changes were calculated by comparing irradiated or H_2_O_2_ treated cells with untreated cells. * = *p* < 0.05. (**B**) Experimental setup for costress model. Cells were treated with NaCl (100 mM or 200 mM) or cultured in normal medium (= Control). Cells were either radiated with UV (10 min 75 W (4500 mJ/cm^2^)), treated with H_2_O_2_ (300 µM), or were untreated 6 h after incubation. Cells were subsequently stained for γH2AX after 3 h of incubation and γH2AX positive cells were identified using image analysis.

## Data Availability

Datasets related to this article can be found at http://www.ncbi.nlm.nih.gov/bioproject/773592, release date 01/07/2022, hosted by the Sequence Read Archive (SRA). Data will become available for the public upon publication.

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
