# Peer review of "Osmotic Stress Interferes with DNA Damage Response and H2AX Phosphorylation in Human Keratinocytes"

_cells, 2022, doi:10.3390/cells11060959_

Round 1

Reviewer 1 Report

The article ''Osmotic stress interferes with DNA damage response and 2
H2AX phosphorylation in human keratinocytes'' by Laura Hoen et al is a well written article that highlights the importance of osmotic stress on DNA damage and its effect on the human keratinocytes. However, there are not enough experiments performed to claim the points made and the results are also not significant.

Author Response

  • We are grateful for the comments of the reviewer and the chance to improve the manuscript.

The reviewer points out that there are not enough experiments performed that is why we decided to add specific qPCR experiments (TaqMan Assays) to confirm the results from the sequencing. Therefore, we will perform the assays for ATM, ATR, PP4 and CCDC6 as already in the Methods section added (line 305ff). We are hopeful that these assays show the same results in a higher significance as seen in the sequencing (see Fig. 3). 

Line 305ff:

“qPCR (TaqMan Assays)

After RNA Isolation, RNA was synthesized to complementary DNA (cDNA) with the High-Capacity RNA-to-cDNA™ Kit (Thermo Fischer, Germany) according to the manufacturer’s protocol. This was performed on a Thermal Cycler T100 (BIO-RAD Labaratories, USA). A uniform RNA concentration was used, starting with the lowest.

Afterwards, the cDNA was used for the TaqMan Assays (Thermo Fischer, Germany). The following assays were done according to the manufacturer’s protocol, each in double determination for each probe:

ATM (Assay ID: Hs00175892_m1), ATR (Assay ID: Hs00992123_m1), PP4 (Assay ID: Hs00908713_m1), CCDC6 (Assay ID: Hs00193731_m1), 18S (Assay ID: Hs99999901_s1)

The assays were performed on a qPCR thermal cycler C1000 with fitting CFX96 Real-Time System (BIO-RAD Laboratories, USA). Analysis were done with Microsoft Excel 2016. First, the Cq values were normalized to the 18S level (Cq of Assay x minus Cq of 18S Assay). The Delta was calculated by using the normalized Cq value of the control (untreated cells), which is subtracted by the normalized Cq value of the treated cells. Afterwards, the DeltaDelta was calculated by the following formula 2^-(Delta). Then the difference of the DeltaDelta of control and treatment was determined for each assay in each probe. A T-Test was performed for statistical significance.”

Reviewer 2 Report

The mansucript entitled “Osmotic stress interferes with DNA damage response and H2AX phosphorylation in human keratinocytes” has been appraised on its priority to be accepted by the Cells journal. The following questions should be addressed before publication in the journal.

Major-

1) Human skin is a multilayer epithelium that is covered with an anucleate stratum corneum to prevent varied stress such as osmotic stress from harmful effects. In general, UVB (290-320nm) can penetrates to the basal layer in where the keratinocyte stem cells reside to cause their DNA damage. The osmotic stress loaden on the outer layer of the skin impossibly directly affect the basal keratinocytes, the authors should give a reasonable interpretation what's impilication point this manuscript places on.

2) Figure 4. The cells were challenged with 200 mM NaCl, 300 microM H2O2, and UV (10 min 75 W) in a very narrow time window. The authors have to give us a panel of treated-cells images that show these cells were survival.

Minor-

1) why does this manuscript submit into the category of “review”, might be the category of original article.

2) the dose of UV (10 min 75 W) should be converted to ? mJ/cm2.

Author Response

Human skin is a multilayer epithelium that is covered with an anucleate stratum corneum to prevent varied stress such as osmotic stress from harmful effects. In general, UVB (290-320nm) can penetrates to the basal layer in where the keratinocyte stem cells reside to cause their DNA damage. The osmotic stress loaden on the outer layer of the skin impossibly directly affect the basal keratinocytes, the authors should give a reasonable interpretation what's implication point this manuscript places on.

UV light penetrates the skin in a wavelength-dependent manner. In the solar radiation, which we used, UVA and UVB are the predominantly occurring lights. Whereas UVB (290-320 nm) is mainly absorbed by the epidermis with also reaching the basal layer, UVA (320-400 nm) can also reach deeply into the dermis. The effect of UVA and UVB on the skin cells is different. UVA efficiently generates reactive oxygen species, while UVB is directly absorbed by DNA, which causes modifications of the DNA. These can lead to mutations and cancer [1].

Keratinocytes can suffer from osmotic stress in two ways. One, called dry skin or xerosis, is caused by excessive water loss of the stratum corneum due to genetical, environmental or aging factors. In this condition the basal keratinocytes are mainly involved through the differentiation process to help form an adequate equilibrium, which is necessary to keep the skin healthy [2]. But when the barrier is breached extracellular water  leaves the skin, which creates a hyperosmotic extracellular milieu in the epidermis [3], which sets the basal keratinocytes under osmotic stress.

In fact, our approach is different from dry skin. We discuss the state of dehydration (in our case hypertonic dehydration with increased sodium concentration in the extracellular fluid [4] (see introduction line 54ff)), which is very common in elderly patients [4,5]. There are empirical evidence suggesting the relationships between fluid intake and skin properties [6,7] and there are hints that additional intake of water seems to influence skin barrier parameter like the stratum corneum hydration [8,9]. As already mentioned in the introduction reduced skin turgor or poor wound healing also reveal a coherence between dehydration status and affected epidermis [10,11] (see line 37ff). As the epidermis is devoid of any blood circulation, their viability relies on bulk diffusion of water and nutrients. But the organic osmolyte transport by keratinocytes is largely unexplored [12]. Nipkey et al. could show that high-salt diet causes hyperosmolality in skin [13] and as Suckling et al. showed, dietary salt increases the plasma sodium concentration [14]. Taken together, due to the elevated sodium concentration in the plasma and extracellular fluid during hypertonic dehydration [4], the water transport from dermis to epidermis is disturbed as the result of hyperosmolality, which sets the keratinocytes under osmotic pressure.

So both conditions, dehydration and dry skin, lead to osmotic stress for keratinocytes. UV light is one factor, which can lead to dry skin [15]. Moreover, dehydration occurs more frequently in summer due to the excessive water loss through sweating and higher transepidermal water loss and in summer human beings are more likely to be exposed to UV. This shows that both stressors, osmotic and UV, are likely to occur together, which underlines the importance of our study. 

2)         Figure 4. The cells were challenged with 200 mM NaCl, 300 microM H2O2, and UV (10 min 75 W) in a very narrow time window. The authors have to give us a panel of treated-cells images that show these cells were survival.

  • Due to the double stress with NaCl and H2O2 / UV, the cells could easily die. That’s why we tried different doses of H2O2 / UV until we figured out one where the cells show a reaction towards the stress, but nearly all cells survive. As a prove, we added another supplementary figure (see line 163 + Supplementary Figure 6), which shows the cell survival of the costress experiments. Nevertheless, cells which undergo both stresses were survival, these cells show in shape and morphology that they are clearly stressed and some of them even show signs of apoptosis, especially when treated with 200 mM NaCl (Fig. S6 G, H). This underlines the aspect that keratinocytes under osmotic pressure can not cope with UV stress (DNA damage) in comparison to untreated (no NaCl) cells (Fig. S6 C), which show no difference when UV light is applied.

Figure S6 – Cell Survival after Costress Experiments. Left side shows the treatment with UV (10 min 75 W), right side shows the treatment with H2O2 (300 µM 10 min); A, B Cells were left untreated (no NaCl, no UV/ H2O2), C, D Cells are treated with UV / H2O2 with no previously NaCl treatment; E, F Cells are treated with 200mM NaCl for 6 hours; G, H Cells are incubated with 200 mM NaCl for 6 hours and treated afterwards with UV / H2O2; I, J Cells are treated with 100mM NaCl for 6 hours; K, L Cells are incubated with 100 mM NaCl for 6 hours and treated afterwards with UV / H2O2

Minor-

1)         why does this manuscript submit into the category of “review”, might be the category of original article.

à Unfortunately, this was a mistake during the submission process. The editors are already informed and it was changed in the manuscript. 

2) the dose of UV (10 min 75 W) should be converted to ? mJ/cm2.

à As suggested by the reviewer, we have added the details in the manuscript. 75 W for 10 min is roughly equivalent to 4500 mJ/cm². We added this to the original settings in brackets (see in line: 185, 328). Since the data in the xenon chamber used can only be made in watts, we have left the original data in the manuscript as a supplement so that it is easier to reproduce the experiment.

Line 180ff:

“Figure 4. – γH2AX identification in costress model. (A) Fold change of γH2AX positive cells after UV radiation (n=4) or H2O2 treatment (n=8) in untreated control cells or cells cultured under osmotic stress conditions prior to radiation or H2O2 treatment. Fold changes were calculated by comparing irradiated or H2O2 treated cells with untreated cells. *=p<0.05 (B) Experimental setup for costress model. Cells were treated with NaCl (100 mM or 200 mM) or cultured in normal medium (=Control). 6 hours after incubation cells were either radiated with UV (10 min 75 W (4500 mJ/cm²)) or treated with H2O2 (300 µM) or were untreated. Cells were subsequently stained for γH2AX after 3 hours of incubation and γH2AX positive cells were identified using image analysis.”

Line 325ff:

Cells were equally seeded in 96-well plates and incubated for 3 days. Afterward, they were either treated with NaCl-enriched medium (100 mM or 200 mM) or given a medium change (control). 6 hours later, the cells were irradiated in a Q-SUN Xe-1 (Q-Lab, Germany) with Daylight Q Filter for 10 min with 75 W (4500 mJ/cm²).”

References

  1. D'Orazio, J.; Jarrett, S.; Amaro-Ortiz, A.; Scott, T. UV radiation and the skin. IJMS 2013, 14, 12222–12248, doi:10.3390/ijms140612222.
  2. Proksch, E.; Berardesca, E.; Misery, L.; Engblom, J.; Bouwstra, J. Dry skin management: practical approach in light of latest research on skin structure and function. J. Dermatolog. Treat. 2020, 31, 716–722, doi:10.1080/09546634.2019.1607024.
  3. El-Chami, C.; Haslam, I.S.; Steward, M.C.; O'Neill, C.A. Role of organic osmolytes in water homoeostasis in skin. Exp. Dermatol. 2014, 23, 534–537, doi:10.1111/exd.12473.
  4. Miller, H.J. Dehydration in the Older Adult. J Gerontol Nurs 2015, 41, 8–13, doi:10.3928/00989134-20150814-02.
  5. Hooper, L.; Bunn, D.; Jimoh, F.O.; Fairweather-Tait, S.J. Water-loss dehydration and aging. Mech. Ageing Dev. 2014, 136-137, 50–58, doi:10.1016/j.mad.2013.11.009.
  6. Megow, I.; Darvin, M.E.; Meinke, M.C.; Lademann, J. A Randomized Controlled Trial of Green Tea Beverages on the in vivo Radical Scavenging Activity in Human Skin. Skin Pharmacol. Physiol. 2017, 30, 225–233, doi:10.1159/000477355.
  7. Jung, S.; Darvin, M.E.; Chung, H.-S.; Jung, B.; Lee, S.-H.; Lenz, K.; Chung, W.-S.; Yu, R.-X.; Patzelt, A.; Lee, B.-N.; et al. Antioxidants in Asian-Korean and caucasian skin: the influence of nutrition and stress. Skin Pharmacol. Physiol. 2014, 27, 293–302, doi:10.1159/000361053.
  8. Palma, L.; Marques, L.T.; Bujan, J.; Rodrigues, L.M. Dietary water affects human skin hydration and biomechanics. Clin. Cosmet. Investig. Dermatol. 2015, 8, 413–421, doi:10.2147/CCID.S86822.
  9. Palma, M.L.; Tavares, L.; Fluhr, J.W.; Bujan, M.J.; Rodrigues, L.M. Positive impact of dietary water on in vivo epidermal water physiology. Skin Res. Technol. 2015, 21, 413–418, doi:10.1111/srt.12208.
  10. Popkin, B.M.; D'Anci, K.E.; Rosenberg, I.H. Water, hydration, and health. Nutrition Reviews 2010, 68, 439–458, doi:10.1111/j.1753-4887.2010.00304.x.
  11. Rikkert, M.G.M.O.; Melis, R.J.F.; Claassen, J.A.H.R. Heat waves and dehydration in the elderly. BMJ 2009, 339, b2663, doi:10.1136/bmj.b2663.
  12. El-Chami, C.; Foster, A.R.; Johnson, C.; Clausen, R.P.; Cornwell, P.; Haslam, I.S.; Steward, M.C.; Watson, R.E.B.; Young, H.S.; O'Neill, C.A. Organic osmolytes increase expression of specific tight junction proteins in skin and alter barrier function in keratinocytes. Br. J. Dermatol. 2021, 184, 482–494, doi:10.1111/bjd.19162.
  13. Nikpey, E.; Karlsen, T.V.; Rakova, N.; Titze, J.M.; Tenstad, O.; Wiig, H. High-Salt Diet Causes Osmotic Gradients and Hyperosmolality in Skin Without Affecting Interstitial Fluid and Lymph. Hypertension 2017, 69, 660–668, doi:10.1161/HYPERTENSIONAHA.116.08539.
  14. Suckling, R.J.; He, F.J.; Markandu, N.D.; MacGregor, G.A. Dietary salt influences postprandial plasma sodium concentration and systolic blood pressure. Kidney Int. 2012, 81, 407–411, doi:10.1038/ki.2011.369.
  15. Geng, R.; Kang, S.-G.; Huang, K.; Tong, T. Boosting the Photoaged Skin: The Potential Role of Dietary Components. Nutrients 2021, 13, doi:10.3390/nu13051691.

Round 2

Reviewer 1 Report

The new data added by the authors significantly improved the overall findings, hence I have provided the I have a favorable recommendation